# Mixed-methods, descriptive and observational cohort study examining feeding and growth patterns among low birthweight infants in India, Malawi and Tanzania: the LIFE study protocol

Linda Vesel ![ORCID],[1] Lauren Spigel,[1] Jnanindra Nath Behera,[2] Roopa M Bellad,[3] Leena Das,[2] Sangappa Dhaded,[3] Shivaprasad S Goudar,[3] Gowdar Guruprasad,[4] Sujata Misra,[2] Sanghamitra Panda,[5] Latha G Shamanur,[6] Sunil S Vernekar,[3] Irving F Hoffman,[7] Tisungane Mvalo,[8,9] Melda Phiri,[8] Friday Saidi ![ORCID],[8] Rodrick Kisenge,[10] Karim Manji,[10] Nahya Salim,[10] Sarah Somji,[10] Christopher R Sudfeld,[11] Linda Adair,[12] Bethany A Caruso,[13] Christopher Duggan,[14] Kiersten Israel-Ballard,[15] Anne CC Lee ![ORCID],[16] Stephanie L Martin,[12] Kimberly L Mansen,[15] Krysten North ![ORCID],[16] Melissa Young,[13] Emily Benotti,[1] Megan Marx Delaney,[1] Eliza Fishman,[1] Katelyn Fleming,[1] Natalie Henrich,[1] Kate Miller,[1] Laura Subramanian,[1] Danielle E Tuller,[1] Katherine EA Semrau ![ORCID] [1,17]

**Correspondence to**
Dr Linda Vesel;
lvesel@ariadnelabs.org

## ABSTRACT

**Introduction** Ending preventable deaths of newborns and children under 5 will not be possible without evidence-based strategies addressing the health and care of low birthweight (LBW, <2.5 kg) infants. The majority of LBW infants are born in low- and middle-income countries (LMICs) and account for more than 60%–80% of newborn deaths. Feeding promotion tailored to meet the nutritional needs of LBW infants in LMICs may serve a crucial role in curbing newborn mortality rates and promoting growth. The Low Birthweight Infant Feeding Exploration (LIFE) study aims to establish foundational knowledge regarding optimal feeding options for LBW infants in low-resource settings throughout infancy.

**Methods and analysis** LIFE is a formative, multisite, observational cohort study involving 12 study facilities in India, Malawi and Tanzania, and using a convergent parallel, mixed-methods design. We assess feeding patterns, growth indicators, morbidity, mortality, child development and health system inputs that facilitate or hinder care and survival of LBW infants.

**Ethics and dissemination** This study was approved by 11 ethics committees in India, Malawi, Tanzania and the USA. The results will be disseminated through peer-reviewed publications and presentations targeting the global and local research, clinical, programme implementation and policy communities.

**Trial registration numbers** NCT04002908 and CTRI/2019/02/017475.

## Strengths and limitations of this study

► Our convergent parallel, mixed-methods design will yield detailed and unique data on supply and demand side aspects of infant care, feeding and growth at the facility and community levels throughout the first year of life.

► The LIFE (Low Birthweight Infant Feeding Exploration) study focuses on low birthweight (LBW) infants with birth weights between 1.5 kg and <2.5 kg given limited data on this birthweight group.

► The multisite approach enables the comparison of results within and across three countries in sub-Saharan Africa and South Asia, where the phenotypes of LBW infants may vary.

► The main limitation of this study is that enrolment is facility-based, missing the population of small infants who are born outside the health facility or delay presentation for care within the health system.

## INTRODUCTION
### Challenge

Although improvements in child health and survival have been achieved in the past two decades, a significant decline in neonatal mortality is needed to attain Sustainable Development Goal 3.2 by 2030.[1] Focusing attention on the most vulnerable small and sick newborns, namely those born low birthweight (LBW, <2.5 kg), can reduce risk and poor outcomes in the first days, weeks and months of life. LBW, resulting from preterm birth and/or intrauterine growth restriction, accounts for 14.6% of newborn births, but represents 60%–80% of newborn deaths.[2] This burden is disproportionately concentrated in

low- and middle-income countries (LMICs).[2] For at least the last 5 years, prematurity has been the predominant cause of mortality for children under 5 years of age.[3] The mortality risk among infants born preterm and small-for-gestational age is more than 15 times greater than those born term and appropriate-for-gestational age.[2 4–7]

Additionally, LBW infants face increased risks for morbidity, neurodevelopmental impairments and growth faltering as well as challenges related to breast feeding, particularly its initiation and exclusivity in early infancy.[2 8–12] Inadequate and insufficient feeding contributes to poor growth outcomes, such as stunting, wasting and underweight, for which the risk among LBW infants is already increased.[13] Without close monitoring of growth and guidance on optimal feeding, LBW infants may not appropriately catch up to their non-LBW peers.

Design, testing and implementation of interventions to optimise feeding, growth and development among LBW infants requires a rigorously investigated foundational understanding of current practices, standard of care, outcomes and resource availability in LMICs.[14] Limited evidence exists, particularly in low-resource settings, on feeding patterns and initiation, the feeding ecosystem, policies for the care of LBW infants and supply and demand inputs related to various feeding modes.[15–17] Generally, studies capturing evidence on optimal milk type and content for moderately LBW infants (1.5 kg to <2.5 kg) in low-resource settings are lacking, hindering the proper management of nutritionally at-risk newborns.[18] The World Health Organization (WHO) feeding guidelines for LBW infants were published in 2011[19]; however, most of the recommendations are based on low quality evidence; even so, the majority of available evidence was from high-income countries, and most data were limited to very LBW infants (<1.5 kg) even though the majority of LBW infants are born between 1.5 kg to <2.5 kg.[12 20]

### Opportunity

Despite these gaps, a number of opportunities in recent years have propelled interest and investment in the care of LBW infants. In 2012, the World Health Assembly highlighted the need to prevent LBW by setting a goal of reducing LBW births by 30% by 2025 as part of a set of 6 nutrition targets comprising its comprehensive implementation plan on maternal, infant and young child nutrition.[2] In 2017, 40 global health organisations issued an urgent call to action for more evidence to address the current state of feeding of sick and vulnerable infants.[21] In the past 2 years, the WHO, the United Nations Children's Fund (UNICEF) and the United States Agency for International Development released reports and guidelines focusing on the care of the small and sick newborn.[22] With interest and investment galvanised, research is needed to develop evidence-based approaches and translate findings into action to ensure the most vulnerable infants can survive and thrive.

### Study aims

The Low Birthweight Infant Feeding Exploration (LIFE) study aims to document current feeding practices and growth patterns among LBW infants in LMICs to inform potential feeding interventions. The formative, observational cohort study has four objectives to be assessed among LBW infants using a mixed-methods approach: (1) understand feeding practices and the standard of care underpinning them; (2) explore the beliefs, facilitators and barriers around the feeding of LBW infants; (3) define and document key longitudinal growth and health outcomes up to 12 months of age; and (4) examine the relationships between infant and maternal characteristics, feeding, growth and child development. Before embarking on the formative research, we will conduct extensive desk reviews to better understand the current LBW infant feeding literature and policies in place. The long-term goal of the LIFE study is to inform the design of a future LBW infant feeding and growth trial and, in turn, strengthen the evidence-base for global infant feeding guidelines.

## METHODS AND ANALYSIS
### Study design

This protocol was developed based on the Strengthening the Reporting of Observational Studies in Epidemiology guidelines, the Consolidated Criteria for Reporting Qualitative Research guidelines, as well as key principles of designing and conducting mixed-methods studies.[23–25] LIFE is a formative, multisite, observational cohort study involving 12 study facilities and using a convergent parallel, mixed-methods design. Quantitative and qualitative data will be collected and analysed in parallel and then merged at the interpretation phase.[25] A mixed-methods approach allows us to establish the comprehensive foundational knowledge to design feeding interventions for nutritionally at-risk LBW infants. The purpose of the quantitative observational descriptive component of the LIFE study (including a retrospective chart review, prospective observational cohort, in-facility observational cohort and facility needs assessments) is to evaluate feeding practices and health outcomes among LBW infants and the health system inputs that support their care. The purpose of the qualitative descriptive component (including in-depth interviews (IDIs) and focus group discussions (FGDs)) is to examine the care and feeding of LBW infants from the perspectives of various key stakeholders. Finally, the integration of the quantitative and qualitative findings will allow us to more fully understand the context and reasons for the feeding patterns and health outcomes that we observe. Further details of the study design, objectives and data collection activities can be found in table 1. Data collection began in August 2019 and is on-going with plans for completion by October 2021.

### Study setting

The LIFE study is implemented in four sites across three countries: (1) Karnataka state, India and (2) Odisha state,

**Table 1** Details of study design, data collection and analysis

| | Quantitative data strand | Qualitative data strand | Merged quantitative and qualitative data strands |
|---|---|---|---|
| Aim | To document current feeding practices and growth patterns among LBW infants in LMICs in order to inform potential feeding interventions | | |
| Objectives | 1. Define and document feeding patterns, and key longitudinal growth and health outcomes from birth to 12 months of age<br>2. Examine the relationships between infant and maternal characteristics, feeding and growth | 1. Explore the beliefs, facilitators and barriers around the feeding of LBW infants | 1. Understand feeding practices and the beliefs, facilitators, barriers and standards of care underpinning them in order to better identify infants at-risk for poor growth and health outcomes |
| Research questions | 1. What are the feeding patterns, growth trajectories and health outcomes among LBW infants from birth to 12 months?<br>2. What are the infant and maternal predictors of poor growth outcomes at 6 and 12 months?<br>3. What are the infant and maternal predictors of non-exclusive breast feeding in the first 6 months?<br>4. What is the association between the duration of exclusive breast feeding and growth outcomes at 6 and 12 months? | 1. What do mothers, family members, community members, healthcare providers and other key stakeholders think LBW infants should be fed and why? | 1. What are the current practices, beliefs, facilitators and barriers regarding the feeding of LBW infants in facility and community settings in LMICs? |
| Study design | Observational, descriptive quantitative data collection and analysis (formative—no intervention) as part of overall convergent parallel design | Descriptive, qualitative data collection and analysis as part of overall convergent parallel design | Convergent parallel design leveraging and merging data from quantitative and qualitative data strands |
| Data collection (activity: sample per site) | ▶ Retrospective chart review: 155 mother–infant pairs<br>▶ Prospective observational cohort: 300 mother–infant pairs<br>▶ In-facility observational cohort: 35 mother–infant pairs<br>▶ Facility needs assessments: 1–5 health facilities | ▶ In-depth interviews: 72 participants<br>▶ Focus group discussions: 12 groups (15–24 participants) | ▶ Quantitative and qualitative data collected in parallel as noted in the respective data strands |
| Data analysis (activity: analysis methods) | ▶ Retrospective chart review: descriptive statistics (means, medians, SD and frequencies)<br>▶ Prospective observational cohort: descriptive statistics (means, medians, SD and frequencies) and models exploring relationships between key characteristics, feeding and growth (t-tests, $\chi^2$ tests and regression - linear, log-binomial, poisson and/or logistic)<br>▶ In-facility observational cohort: descriptive statistics (means, medians, SD and frequencies)<br>▶ Facility needs assessment: descriptive statistics (means, medians, SD and frequencies) | ▶ In-depth interviews: thematic framework analysis<br>▶ Focus group discussions: thematic framework analysis | ▶ Quantitative and qualitative data analysed in parallel |
| Interpretation | Merging of findings from quantitative and qualitative data collection strands to compare and contrast findings and provide recommendations on optimal feeding options and timing of growth monitoring in order to prevent infants from becoming nutritionally at-risk | | |

LBW, low birthweight; LMICs, low-income and middle-income countries.

**Table 2** Site descriptions

| Site | Prevalence of LBW[2 47] | Neonatal mortality rate[48] Deaths per 1000 live births | Infant mortality rate[49] Deaths per 1000 live births | Study facilities Number and type by site |
|---|---|---|---|---|
| India—Karnataka | 17.2% | 22 | 28 | Three private tertiary hospitals Two public tertiary hospitals |
| India—Odisha | 20.8% | | | One public tertiary hospital One public secondary hospital |
| Malawi | 14.5% | 20 | 31 | One public tertiary hospital One public secondary hospital |
| Tanzania | 10.5% | 20 | 36 | One public tertiary hospital Two public secondary hospitals |

LBW, low birthweight.

India, led by teams at Jawaharlal Nehru Medical College, JJM Medical College, SS Institute of Medical Sciences, City Hospital and Srirama Chandra Bhanja Medical College; (3) Lilongwe, Malawi, led by the team at University of North Carolina (UNC) Project Malawi; and (4) Dar es Salaam, Tanzania, led by the team at Muhimbili University of Health and Allied Sciences. All sites have strong, long-standing relationships with key government stakeholders, placing them in a position to address local and national priorities and advocate for translation of research to practice. Sites were chosen in South Asia and sub-Saharan Africa as these regions represent the greatest burden of LBW as well as the diverse drivers of LBW.[2] Investigators will document regional similarities and differences in maternal and infant characteristics, gestational age, birthweight, feeding patterns and growth. In total, participants are recruited from 12 study facilities (ie, secondary and tertiary hospitals) chosen based on delivery volume, capacity to care for LBW infants in the first days of life and willingness of facility leadership to participate. All facilities are located in urban settings with participants residing within a 50 km radius. Additional site details are included in table 2.

### Study population
#### Quantitative
For the quantitative data collection activities, the study population comprises mother–infant pairs that include newborns with recorded birth weights of 1.5 kg to <2.5 kg, as well as health facilities (table 3). All 12 study facilities will participate in the facility needs assessments, including a facility profile and a donor human milk (DHM) bank assessment; further inclusion and exclusion criteria have not been specified.

#### Qualitative
For the qualitative data collection activities, the study population includes mothers, family members (eg, husbands, guardians, mothers' parents, mother's in-laws, grandmothers, sisters and sisters-in-law), religious and community leaders, traditional healers, clinicians, government officials, supply chain experts and DHM banking experts meeting specific inclusion criteria (table 3). Mothers and family members of newborns with recorded birth weights of 1.5 kg to <2.5 kg are eligible for the IDIs and FGDs. Clinicians who have been at their position for less than 6 months are not eligible.

### Patient and public involvement
As part of the study design, the LIFE team involves clinicians, researchers and community stakeholders familiar with the respective settings and populations. Study tools were piloted with patients and community members to ensure that research questions and indicators are culturally appropriate, acceptable and relevant to the study population.

### Study measures
#### Quantitative
We evaluate key maternal and infant characteristics (eg, maternal education, maternal age, place of residence, parity, place and type of delivery, gestational age, infant sex and average length of facility stay) to better understand the LBW population in each site and region and to evaluate which characteristics serve as predictors of particular feeding patterns and growth and health outcomes. We examine infant feeding patterns including: early initiation of breast feeding (within 1 hour of birth), feeding profiles (exclusive breastmilk, mixed milk feeding or no breastmilk) at each visit week from birth to 12 months and duration of exclusive breast feeding (feeding of only breastmilk directly from the breast, expressed or from a donor). We will assess infant growth by measuring weight, length, head circumference and mid-upper arm circumference (MUAC) at 13 time points. Anthropometrics will be used to identify stunting (length-for-age

**Table 3** Inclusion and exclusion criteria for mothers–infant pairs

| Data collection activity | Inclusion criteria | Exclusion criteria |
|---|---|---|
| Retrospective chart review | ► Infants with birthweight of 1.5 kg to <2.5 kg.<br>► Infants discharged before the start of prospective data collection for LIFE study. | ► Infants with birth weight <1.5 kg.<br>► Infants with congenital abnormalities that interfere with feeding (cleft lip or palate; hydrocephalus; gastrointestinal tract anomalies including gastroschisis, omphalocele or anal atresia; neural tube defects; congenital cardiac defects; suspected trisomy 21; suspected TORCH (Toxoplasmosis, Other agents, Rubella, Cytomegalovirus and Herpes simplex) infection.<br>► Infants with young mothers: <18 years old in Tanzania and India, 16–17 years old and unmarried in Malawi and all mothers <16 years old in Malawi.<br>► Infants who die less than 72 hours from the time of birth.<br>► Infants born outside the facility. |
| Longitudinal prospective cohort | ► Infants with birthweight between 1.5 kg to <2.5 kg (as measured at birth, or calculated using algorithm based on time since birth to account for expected postnatal weight loss).<br>► Mother–infant pairs who reside within the catchment area (approximately 50 km) of the facility in which they were enrolled. | ► Infants with birth weight <1.5 kg.<br>► Infants with congenital abnormalities that interfere with feeding (cleft lip or palate; hydrocephalus; gastrointestinal tract anomalies including gastroschisis, omphalocele or anal atresia; neural tube defects; congenital cardiac defects; suspected trisomy 21; suspected TORCH infection. Infants with severe neonatal encephalopathy jeopardising early survival (as determined by modified Sarnat criteria).[50 51]<br>► Infants with young mothers: <18 years old in Tanzania and India, 16–17 years old and unmarried in Malawi and all mothers <16 years old in Malawi.<br>► Infants with mothers who died prior to enrolment.<br>► Infants who die less than 72 hours from the time of birth.<br>► Infants older than 72 hours at the time of screening.<br>► Infants who withdraw less than 72 hours from the time of birth.<br>► Infants with a twin or triplet who die prior to the time of screening.<br>► Mothers who plan to leave the catchment area within 6 months of study enrolment. |
| In-facility observational cohort | ► Infants with birthweight between 1.5 kg to <2.5 kg. | ► Infants with birth weight <1.5 kg.<br>► Infants with congenital abnormalities that interfere with feeding.<br>► Infants with young mothers: <18 years old in Tanzania and India, 16–17 years old and unmarried in Malawi and all mothers <16 years old in Malawi.<br>► Infants who die less than 6 hours from the time of birth.<br>► Infants with mothers who die less than 6 hours from the time of birth.<br>► Infants born outside the facility.<br>► Infants older than 6 hours at the time of screening. |

Continued

**Table 3** Continued

| Data collection activity | Inclusion criteria | Exclusion criteria |
|---|---|---|
| In-depth interviews and focus group discussions (birth—6 months) | ► Mothers with infants with birthweight between 1.5 kg to <2.5 kg aged 0–7 months (enrolment in prospective observational cohort not required).<br>► Family members of infants with birthweight between 1.5 kg to <2.5 kg aged 0–7 months who play a role in infant and young child feeding.<br>► Religious leaders, community leaders and traditional healers that are opinion leaders on infant and young child feeding practices in the community.<br>► Healthcare workers currently involved in providing infant and young child feeding.<br>► Government officials who support infant and young child feeding programmes and policies.<br>► Supply chain experts involved in infant and young child feeding supply chain logistics.<br>► Human milk bank experts. | ► Mothers and family members with infants with birth weight ≥2.5 kg.<br>► Mothers and family members with infants with birth weight <1.5 kg.<br>► Young mothers: <18 years old in Tanzania and India, 16–17 years old and unmarried in Malawi and all mothers <16 years old in Malawi.<br>► Healthcare workers who have been in their position for less than 6 months.<br>► Government officials who have been at their post for less than 6 months. |
| In-depth interviews (9–12 months) | ► Mothers with infants with birthweight between 1.5 kg to <2.5 kg enrolled in the prospective observational cohort and aged 9–12 months. | ► Mothers with infants who were not enrolled in the prospective cohort or withdrew/died before 9 months of age. |

LIFE, Low Birthweight Infant Feeding Exploration.

z-score <-2SD), wasting (weight-for-length z-score <-2SD) and underweight (weight-for-age z-score <-2SD) at 6 and 12 months; and plot growth trajectories and velocities. Z-scores are derived from the International Fetal and Newborn Growth Consortium for the 21st Century (INTERGROWTH-21st) newborn size at birth and preterm postnatal growth standards[26 27] and the WHO infant growth standards[28] from birth through 12 months. We also examine weeks to birthweight regain, namely lack of regain in the first 2 weeks[29–31]; and identify infants experiencing slow weight gain (<20 g/day) in the second week of life (infants are meant to gain an average of 20 g/day in the first month of life and the second week of life is when infants should be gaining back their postnatal weight loss).[32 33] The WHO standards were designed using a cohort of term infants while the INTERGROWTH-21st standards were designed specifically for preterm infants and serve as a complement to the WHO standards in the first 6 months of life. Additional health outcomes include infant morbidity based on maternal self-report of illnesses and symptoms experienced by the infants in the past week at each study visit; neonatal and infant mortality at any point during follow-up; and child development assessed via the Caregiver Reported Early Childhood Development Instruments at 1 year of age.[34] Maternal study measures including illness, depression (based on the Patient Health Questionnaire-2) and anthropometrics (weight, height and MUAC taken when the infant was born, 6 weeks, 6 months and 12 months) are collected to evaluate mothers' roles in the feeding and growth of their infants.[35] Finally, data on bed capacity, length of stay post-delivery, neonatal intensive care admissions, infrastructure, space, equipment, feeding options, medications, staffing, service availability and sanitation practices are gathered to assess the health facility inputs available for care provision of LBW infants.

### Qualitative
We explore beliefs, barriers, facilitators, risks and benefits of various feeding practices through structured IDIs and FGDs.

### Data collection
Quantitative and qualitative data collection occur in parallel through 6 descriptive study activities (table 1). For most data collection activities, timing of data collection will be linked with infant age (figure 1). All data collection is prospective except for the chart reviews. Facility needs assessments are not included in figure 1 since they will not be linked to infant age; facility profiles are completed at baseline and DHM assessments over the course of the study.

### Quantitative
Quantitative data collection is conducted by trained research nurses. The use of multiple data collection activities, combining observations and maternal self-report, helps to reduce bias and allows for triangulation of data. Retrospective patient chart data was collected for infants born in 12 study facilities between July 2018 and October 2019 using a structured survey. Chart reviews were completed before the initiation of prospective observational cohort data collection. The goal for the prospective cohort is to include 300 mother–infant pairs per site

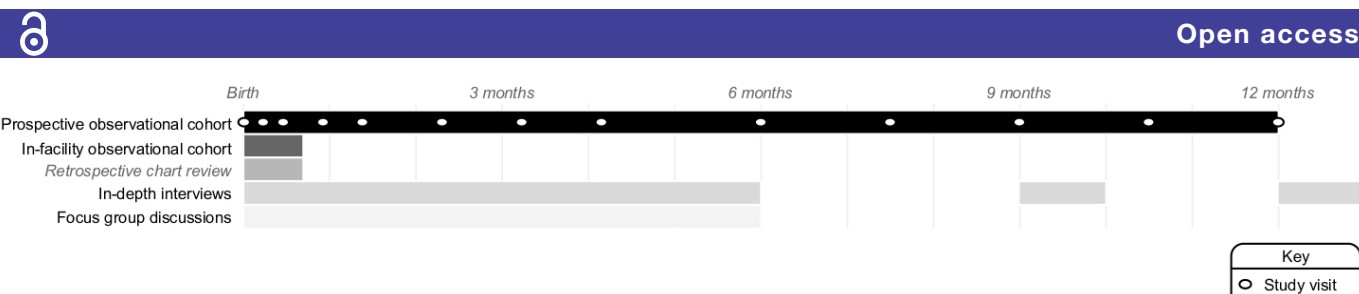

**Figure 1**  Data collection timeline by activity and infant age.

during 13 study visits using structured survey tools for infants born between September 2019 and January 2020 (noting that sites will start and end at different times) and completing 1-year follow-up in July 2021 (figure 1). Data are collected in-person at all time points, apart from at 32.5 and 45.5 weeks, at which time an abbreviated survey is administered over the phone in Malawi and Tanzania to bridge the 3-month time gap between in-person visits in order to reduce loss to follow-up; sites in India have previous experience with at least 3-month intervals between visits without adverse consequences on follow-up rates. Changes made in response to the COVID-19 pandemic included the addition of COVID-19 symptom screening questions, pauses to enrolment, data collection via phone calls where in-person visits were not possible and widening of the 6 month visit window. Table 4 summarises the assessments and timing of administration for the prospective observational cohort. Focused training was conducted by a paediatrician or research investigator for all assessments listed. Although this is an observational cohort study with no intervention, adverse events are monitored and recorded. The in-facility observational cohort is intended to include 35 mother–infant pairs per site from all participating study facilities via regular

**Table 4**  Timing of assessments for longitudinal prospective observational cohort

| Assessment or survey tool content | Age of infant (weeks) | | | | | | | | | | | | |
|---|---|---|---|---|---|---|---|---|---|---|---|---|---|
| | 0 | 1 | 2 | 4 | 6 | 10 | 14 | 18 | 26 | 32.5 | 39 | 45.5 | 52 |
| Maternal demographics and pregnancy history* | • | | | | | | | | | | | | |
| Infant demographics and delivery information* | • | | | | | | | | | | | | |
| Dubowitz examination for gestational age at birth[52 53] | • | | | | | | | | | | | | |
| Infant anthropometrics* | • | • | • | • | • | • | • | • | • | | • | | • |
| Maternal anthropometrics | • | | | | • | | | | • | | | | • |
| Maternal and infant health information | • | • | • | • | • | • | • | • | • | | • | | • |
| Maternal and infant mortality information* | • | • | • | • | • | • | • | • | • | • | • | • | • |
| The WHO-5 Well-Being Index[54] | | | | | | | | | | | • | | • |
| Maternal lactation and infant feeding information* | • | • | • | • | • | • | • | • | • | • | • | • | • |
| Infant and Young Child Feeding Questionnaire for complementary feeding period[55] | | | | | | | | | | | • | | • |
| Latch, Audible Swallowing, Nipple Type, Comfort and Hold breastfeeding assessment*[56] | | • | • | • | • | • | • | • | • | | | | |
| Preterm Infant Breastfeeding Behaviour Scale*[57] | | • | • | • | • | | | | | | | | |
| Neonatal Eating Assessment Tool[58] | | • | • | • | • | • | • | • | • | | | | |
| Water, sanitation and hygiene information* | | • | • | • | • | • | • | • | • | | • | | • |
| Patient Health Questionnaire 2 on maternal depression[59] | | • | • | • | • | • | • | • | • | | • | | • |
| Caregiver Reported Early Childhood Development Instrument[34] | | | | | | | | | | | | | • |

*Assessments also to be completed for the in-facility observational cohort between birth and facility discharge.

feeding observations (15 min each) and maternal reports starting within 6 hours of birth and continuing until facility discharge (first week—every 3–4 hours; second week—one to two times a day; and third week and thereafter—one time a day for unstable infants and every 3 days for stable infants) (table 4). Finally, facility needs assessment data are collected via standardised tools capturing key vital statistics; the structural, human resource, equipment and service inputs present for new mothers and newborns; and facility and programmatic readiness for establishing and strengthening DHM banks.[29–32] Study team members administer the assessments at each of the study facilities through observations, record reviews and staff consultations; global DHM experts will participate in and provide detailed guidance for the DHM component.

All sample sizes were determined to account for timeline, feasibility and resource constraints. Apart from the prospective observational cohort, we will not aim to make statistical inferences or precise point estimates from these samples, but will use the data descriptively for each infant to construct a narrative of their feeding patterns and health outcomes. The main statistical results from the prospective observational cohort will be point estimates and CIs for certain rates, such as the per cent of LBW infants who fail to thrive or whose growth falters, develop problems breast feeding or fall ill. The size of the true proportions will determine the precision of the estimate (eg, with a sample of 300 mother–infant pairs at each site, a true proportion of 10% can be detected with precision of ±3.6%). The 95% CI would be 6.4% to 13.6% (figure 2).

To ensure standardisation and facilitate high quality data collection, investigators will conduct site-specific workshops to review tools and train research staff on how to prepare for and conduct all key assessments; timing and duration will be tailored to the needs and knowledge of each team. Supervisors perform regular quality checks and refresher training as needed. Standardised anthropometric equipment includes: Seca 334 mobile digital baby scale, 887/876/874 digital flat scale with foot pedal for maternal weight, site-specific height boards for mothers, Seca 417 infant measuring board, Shorr MUAC tapes for infants and mothers (WM-MUAC26) and Shorr 65 cm head circumference tapes for infants (SKU: WM-S Tape).

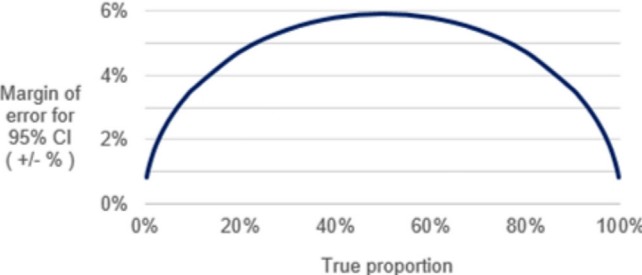

**Figure 2** Margin of error in estimation of a single proportion with a sample of 300 at each study site for the prospective observational cohort.

The survey tools are administered in the local language (Hindi, Kannada and Marathi in India—Karnataka; Hindi and Oriya in India—Odisha; Chichewa in Malawi; and Swahili in Tanzania).

### Qualitative
IDIs and FGDs with mothers and family members of LBW infants, community members, clinicians, government officials and supply chain and DHM knowledge experts took place between July 2019 and January 2021. All participants will only be interviewed once. Interviewers include men and women with clinical and/or research backgrounds including specialised training in qualitative data collection methods. They conduct IDIs and FGDs with stakeholders and clinicians in English, and with mothers, family members and community leaders in relevant local languages. Interviewers are not personally known to interviewees and are trained to reduce bias by building rapport with all participants and maintaining neutrality and confidentiality. They start each IDI and FGD by sharing the purpose of the study and any potential risks, obtaining consent and then use structured interview guides with specific probes; guides were piloted prior to data collection. All IDIs and FDGs are audio-recorded with permission from participants and supported with real-time note-taking; transcripts and/or summary notes will not be reviewed by participants. IDIs are scheduled for 1 hour and FGDs for 2 hours. All qualitative data collection takes place in a private location either within a study health facility or in the community setting. Participants are given a small incentive to participate in order to cover the cost of travel and lost wages.

We use purposive sampling for government officials and knowledge experts and convenience sampling based on availability for clinicians, mothers, family members and community leaders. Participants are approached in person or via email and visits are confirmed over the phone. Mothers of LBW infants aged 0–3 months and 4–7 months are sampled from facility patient charts while those aged 9 and 12 months are sampled from the prospective observational cohort based on their sex and presence or absence of stunting at 6 months of age. Our sampling strategy is designed to reach code saturation rather than meaning saturation.[36]

### Data management and analysis
#### Quantitative
Quantitative data are entered and managed in CommCare,[37] a secure electronic data capture and management system (retrospective chart reviews and prospective observational cohorts) and Microsoft Excel 2018, version 16.16.27 (facility needs assessments). For the prospective observational cohort only, the Tanzania team uses its own electronic data capture system and securely transfers data to be merged with the rest of the data collected in CommCare. All data used for analysis are de-identified and stored securely. Experienced statistical

analysts and epidemiologists will use means, medians and SD to describe continuous variables and frequencies and proportions for categorical variables (all activities). For the prospective observational cohort, t-tests, $\chi^2$ tests and regression models (linear, logistic, log-binomial and/or poisson) will be used to explore relationships between key maternal and infant characteristics, feeding and growth. CIs around all measures will be constructed to adjust by site and cluster by mother to account for multiple births. Analyses will be stratified by site, sex and LBW phenotype. Missing data will be accounted for and denominators will be adjusted accordingly. Outliers for z-scores and anthropometrics will be examined and cleaned with the use of existing guidelines and sensitivity analyses, where helpful. Any further analyses will be post hoc and described separately. Analyses will be conducted using the Stata Statistical Software Package, V.16[38] and the SAS statistical software suite.[39]

## Qualitative

De-identified qualitative data are stored for internal use in SharePoint, a secure web-based document management and storage system. A codebook will be developed deductively (based on the research aims and interview guide questions to inform high-level codes) and inductively (based on the emerging content of the IDIs and FGDs to inform subcoded and emerging high-level codes) and applied to IDI and FGD data. Coding of the IDIs and FGDs will be performed by a total of seven coders and will involve a combination of rapid (framework analysis based on real-time notes supported by audio-recordings, where needed) and in-depth (coding of verbatim translated transcripts) approaches.[40–42] Interviewers/site-based coders will employ a framework analysis and code notes into summary tables followed by subsequent coding (using a codebook) in Dedoose by qualitative researchers at Harvard. The first stage of data collection and coding in India and Malawi will be used to enable ongoing learning during the data collection process for a large volume of qualitative data.[39] For maternal IDIs conducted among those with infants ages 9 months and older in India and Malawi, as well as all of Tanzania's IDIs and FGDs across the full year of infancy, interviewers will transcribe the audio-recordings verbatim in English and the qualitative researchers at Harvard will conduct a comprehensive, thematic analysis of the data to identify key messages as well as similarities and differences across sites. During the analysis, researchers at Harvard will review all coded data and will identify, via Dedoose, themes that were most and least commonly mentioned. Themes will be further defined and discussed with the site-based coders to review the interpretation of the data and reduce bias. Data will be analysed by participant type and study location, and emergent key themes will be used to inform recommendations for future interventions.

## Protocol and registration

The study is registered with ClinicalTrials.gov and Clinical Trial Registry of India (http://ctri.nic.in).

## ETHICS AND DISSEMINATION

This study was approved by 11 ethics committees in India, Malawi, Tanzania and the USA: (1) India Health Ministry's Screening Committee with Indian Council of Medical Research acting as its secretariat (2019–2674); (2) Directorate of Health and Family Welfare Services, Government of Karnataka, which also covers investigators at Women and Children Hospital, Davangere and Chigateri General District Hospital, Davangere (NHM/SPM/04/2019–20); (3) Institutional Ethics Committee of KLE Academy of Higher Education and Research which also covers investigators at JN Medical College, Belagavi and KLES Dr Prabhakar Kore Hospital & Medical Research Center, Belagavi (KAHER/IEC/2019–20/D-2760); (4) Institutional Ethics Review Board of SS Institute of Medical Sciences and Research Centre (IERB/200/2019); (5) Institutional Ethics Committee of JJM Medical College (JJMMC/IEC-01/2019), which also covers investigators at Bapuji Child Health Institute and Research Centre, Davangere, Women and Children Hospital, Davangere and Chigateri General District Hospital, Davangere; (6) Research and Ethics Committee, Directorate of Health Services, Odisha state, which also covers investigators at City Hospital Oriya Bazar, Cuttack (155/PMU/187/17); (7) Institutional Ethical Committee, Sriram Chandra Bhanja Medical College, Cuttack (7188); (8) Malawi National Health Sciences Research Committee (NHSRC2019/Protocol19/03/2250-UNCPM 21905); (9) Tanzania National Institute of Medical Research (NIMR/HQ/R.8a/Vol.IX/3126); (10) Muhimbili University of Health and Allied Sciences (DA.282/298/01.C/); and (11) Harvard T.H Chan School of Public Health (IRB10-0282) which also covers investigators at Boston Children's Hospital, Brigham and Women's Hospital, Emory University, PATH and University of North Carolina.

Written informed consent is obtained from all IDI, FGD, in-facility observational cohort and prospective observational cohort participants. After 6 months of follow-up, prospective observational cohort participants will be re-consented for an additional 6 months. Verbal consent will only be obtained in the event of extenuating circumstances where written consent cannot be sought; Institutional Review Board approval will be needed before proceeding. For the facility needs assessments, verbal consent is sought at the facility leadership level as these activities are considered to be a part of quality improvement. Results from the LIFE study will be disseminated at global-levels and site-levels through peer-reviewed publications and presentations to key stakeholders during meetings and conferences. Site investigators will also share results informally with participating study facilities. On publication of the study results, select data will be made publicly available via Harvard Dataverse.[43]

## DISCUSSION

The LIFE study will fill critical data gaps in the care and nutrition of LBW infants in LMICs; although limited, most of the existing research in this area is concentrated in high-income countries.[18] Overall, evidence is lacking on feeding, care and health of LBW infants born with birth weights of 1.5 to <2.5 kg; however, these moderately LBW infants represent the majority (>90%) of global LBW births compared with very LBW infants (<1.5 kg).[2 17 18 44–46] We will aim to establish the foundational knowledge required to design, test and implement the most effective and feasible infant feeding strategies to prevent and address growth faltering among LBW infants in low-resource settings. The LIFE study will provide the much-needed evidence to comprehensively understand what LBW infants require, how they differ from infants born ≥2.5 kg and what rigorous research is critical to strengthen global LBW infant feeding guidelines.

**Author affiliations**
[1]Ariadne Labs, Harvard T.H. Chan School of Public Health / Brigham and Women's Hospital, Boston, Massachusetts, USA
[2]Department of Paediatrics, SCB Medical College and Hospital, Cuttack, Orissa, India
[3]Jawaharlal Nehru Medical College, KLE Academy of Higher Education and Research, Belgaum, Karnataka, India
[4]Department of Neonatology, JJM Medical College, Davangere, Karnataka, India
[5]Department of Obstetrics and Gynaecology, City Hospital, Cuttack, Orissa, India
[6]Department of Paediatrics, SS Institute of Medical Sciences and Research Center, Davangere, Karnataka, India
[7]Institute for Global Health and Infectious Diseases, University of North Carolina at Chapel Hill School of Medicine, Chapel Hill, North Carolina, USA
[8]Department of Pediatrics, University of North Carolina Project Malawi, Lilongwe, Malawi
[9]Department of Pediatrics, University of North Carolina at Chapel Hill School of Medicine, Chapel Hill, North Carolina, USA
[10]Department of Pediatrics and Child Health, Muhimbili University of Health and Allied Sciences, Dar es Salaam, Tanzania
[11]Department of Global Health and Population, Harvard T.H. Chan School of Public Health, Boston, Massachusetts, USA
[12]Department of Nutrition, University of North Carolina at Chapel Hill Gillings School of Global Public Health, Chapel Hill, North Carolina, USA
[13]Hubert Department of Global Health, Emory University School of Public Health, Atlanta, Georgia, USA
[14]Center for Nutrition, Boston Children's Hospital, Boston, Massachusetts, USA
[15]Maternal, Newborn, Child Health and Nutrition Program, PATH, Seattle, Washington, USA
[16]Department of Pediatric Newborn Medicine, Brigham and Women's Hospital, Boston, Massachusetts, USA
[17]Department of Medicine, Harvard Medical School, Boston, Massachusetts, USA

**Acknowledgements** The authors would like to thank clinical leadership and staff at all study facilities for their partnership, support and contribution to this work; the mothers and infants for allowing us to have a glimpse into their experiences and sharing key moments of their lives; community members, government officials and subject experts for sharing their perspectives; and all data collectors and study staff for conducting study activities.

**Contributors** Study conceptualisation and design was completed by ACCL, BAC, CD, CRS, DET, EB, IFH, KEAS, KI-B, KLM, KMa, KMi, KN, LA, LSp, LSu, LV, MY, NH, RMB, SD, SSG and TM. Protocol development and tool design was carried out by ACCL, BAC, CRS, DET, EB, EF, GG, IFH, JNB, KEAS, KF, KI-B, KLM, KMa, KMi, KN, LA, LD, LGS, LSp, LSu, LV, MY, MMD, NH, NS, RK, RMB, SLM, SM, SP, SMD, SSG, SS, SSV and TM. Tool pilot testing/modification were conducted by EB, EF, FS, GG, JNB, KF, LD, LGS, LSp, LSu, LV, MP, NH, NS, SM, SP, SS and SSV. Writing of the

in low birthweight infants from rural Haryana, India: findings from a secondary data analysis. *BMJ Open* 2018;8:e020384.

12 Rana R, McGrath M, Gupta P, *et al*. Feeding interventions for infants with growth failure in the first six months of life: a systematic review. *Nutrients* 2020;12. doi:10.3390/nu12072044. [Epub ahead of print: 09 Jul 2020].

13 Christian P, Lee SE, Donahue Angel M, *et al*. Risk of childhood undernutrition related to small-for-gestational age and preterm birth in low- and middle-income countries. *Int J Epidemiol* 2013;42:1340–55.

14 Lawn JE, Kerber K, Enweronu-Laryea C, *et al*. Newborn survival in low resource settings--are we delivering? *BJOG* 2009;116 Suppl 1:49–59.

15 Blanc AK, Wardlaw T. Monitoring low birth weight: an evaluation of international estimates and an updated estimation procedure. *Bull World Health Organ* 2005;83:178–85.

16 Gisore P, Shipala E, Otieno K, *et al*. Community based weighing of newborns and use of mobile phones by village elders in rural settings in Kenya: a decentralised approach to health care provision. *BMC Pregnancy Childbirth* 2012;12:15.

17 World Health Organization, United Nations Children's Fund (UNICEF). Low birthweight: country, regional and global estimate World Health Organization; 2004. https://apps.who.int/iris/handle/10665/43184

18 North K, Marx Delaney M, Bose C, *et al*. The effect of milk type and fortification on the growth of low-birthweight infants: an umbrella review of systematic reviews and meta-analyses. *Matern Child Nutr* 2021;17:e13176.

19 World Health Organization. *Guidelines on optimal feeding of low birth-weight infants in low- and middle-income countries*. Geneva, 2011.

20 Brown S, McSharry P. Improving accuracy and usability of growth charts: case study in Rwanda. *BMJ Open* 2016;6:e009046.

21 Urgent call for an investigation into the feeding of sick & vulnerable newborns | by JustACTIONS | Medium [Internet]. Available: https://medium.com/@JustACTIONS/urgent-call-for-an-investigation-into-the-feeding-of-sick-vulnerable-newborns-3f4ddb49c494 [Accessed 24 Jun 2021].

22 WHO. *Survive and thrive: transforming care for every small and sick newborn*. Geneva, Switzerland: World Health Organization, 2018.

23 Strengthening the reporting of observational studies in epidemiology (STROBE) statement: guidelines for reporting observational studies. *BMJ* 2007;335.

24 Tong A, Sainsbury P, Craig J. Consolidated criteria for reporting qualitative research (COREQ): a 32-item checklist for interviews and focus groups. *Int J Qual Health Care* 2007;19:349–57.

25 Creswell JW, Vicki L, Clark P. *Designing and conducting mixed methods research*. 3rd ed, 2017.

26 Villar J, Giuliani F, Bhutta ZA, *et al*. Postnatal growth standards for preterm infants: the Preterm Postnatal Follow-up Study of the INTERGROWTH-21(st) Project. *Lancet Glob Health* 2015;3:e681–91.

27 Villar J, Cheikh Ismail L, Victora CG, *et al*. International standards for newborn weight, length, and head circumference by gestational age and sex: the newborn cross-sectional study of the INTERGROWTH-21st project. *Lancet* 2014;384:857–68.

28 de Onis M, Onyango AW, Borghi E, *et al*. Comparison of the world Health organization (who) child growth standards and the National center for health Statistics/WHO international growth reference: implications for child health programmes. *Public Health Nutr* 2006;9:942–7.

29 Noel-Weiss J, Courant G, Woodend AK. Physiological weight loss in the breastfed neonate: a systematic review. *Open Med* 2008;2:e99–110.

30 Macdonald PD, Ross SRM, Grant L, *et al*. Neonatal weight loss in breast and formula fed infants. *Arch Dis Child Fetal Neonatal Ed* 2003;88:472F–6.

31 DiTomasso D, Cloud M. Systematic review of expected weight changes after birth for full-term, breastfed newborns. *J Obstet Gynecol Neonatal Nurs* 2019;48:593–603.

32 Fenton TR, Anderson D, Groh-Wargo S, *et al*. An attempt to standardize the calculation of growth velocity of preterm Infants-Evaluation of practical bedside methods. *J Pediatr* 2018;196:77–83.

33 Homan GJ. Failure to thrive: a practical guide. *Am Fam Physician* 2016;94:295–9.

34 McCoy DC, Sudfeld CR, Bellinger DC, *et al*. Development and validation of an early childhood development scale for use in low-resourced settings. *Popul Health Metr* 2017;15:3.

35 Kroenke K, Spitzer RL, Williams JBW. The patient health Questionnaire-2: validity of a two-item depression screener. *Med Care* 2003;41:1284–92.

36 Hennink MM, Kaiser BN, Marconi VC. Code saturation versus meaning saturation: how many interviews are enough? *Qual Health Res* 2017;27:591–608.

37 Dimagi. *CommCare*. Cambridge, MA: Dimagi, 2021.

38 StataCorp LLC. *Stata statistical software*. College Station, TX: StataCorp LLC, 2019.

39 SAS Institute. *Sas software suite*. Cary, NC: SAS Institute, 2020.

40 Gale RC, Wu J, Erhardt T, *et al*. Comparison of rapid vs in-depth qualitative analytic methods from a process evaluation of academic detailing in the Veterans health administration. *Implement Sci* 2019;14:11.

41 Taylor B, Henshall C, Kenyon S, *et al*. Can rapid approaches to qualitative analysis deliver timely, valid findings to clinical leaders? a mixed methods study comparing rapid and thematic analysis. *BMJ Open* 2018;8:e019993.

42 Gale NK, Heath G, Cameron E, *et al*. Using the framework method for the analysis of qualitative data in multi-disciplinary health research. *BMC Med Res Methodol* 2013;13:117.

43 Harvard Dataverse [Internet]. Available: https://dataverse.harvard.edu/ [Accessed 24 Jun 2021].

44 O'Leary M, Edmond K, Floyd S, *et al*. A cohort study of low birth weight and health outcomes in the first year of life, Ghana. *Bull World Health Organ* 2017;95:574–83.

45 Mahumud RA, Sultana M, Sarker AR. Distribution and determinants of low birth weight in developing countries. *J Prev Med Public Health* 2017;50:18–28.

46 Marete I, Ekhaguere O, Bann CM, *et al*. Regional trends in birth weight in low- and middle-income countries 2013-2018. *Reprod Health* 2020;17:176.

47 IIPS. *National family health survey 2015-2016*. Mumbai: International Institute for Population Sciences, 2017.

48 Countdown Country Profiles [Internet]. Countdown to 2030. Available: https://profiles.countdown2030.org/#/ds/MWI [Accessed 30 Oct 2020].

49 IGME. IGME UN Inter-agency Group for Child Mortality Estimation [Internet], 2020. Available: https://childmortality.org/ [Accessed 07 Dec 2020].

50 Sarnat HB, Sarnat MS. Neonatal encephalopathy following fetal distress. A clinical and electroencephalographic study. *Arch Neurol* 1976;33:696–705.

51 Roland EH, Hill A. Clinical aspects of perinatal hypoxic-ischemic brain injury. *Semin Pediatr Neurol* 1995;2:57–71.

52 Clopton N. The Dubowitz assessment of gestational age. *Phys Occup Ther Pediatr* 1983;3:75–86.

53 Dubowitz LM, Dubowitz V, Goldberg C. Clinical assessment of gestational age in the newborn infant. *J Pediatr* 1970;77:1–10.

54 Topp CW, Østergaard SD, Søndergaard S, *et al*. The WHO-5 well-being index: a systematic review of the literature. *Psychother Psychosom* 2015;84:167–76.

55 Indicators for assessing infant and young child feeding practices: definitions and measurement methods. Geneva: World Health Organization and the United Nations Children's Fund (UNICEF). Licence: CC BYNC-SA 3.0 IGO, 2021. Available: https://www.who.int/publications/i/item/9789240018389 [Accessed 11 Nov 2021].

56 Jensen D, Wallace S, Kelsay P. LATCH: a breastfeeding charting system and documentation tool. *J Obstet Gynecol Neonatal Nurs* 1994;23:27–32.

57 Hedberg Nyqvist K, Ewald U. Infant and maternal factors in the development of breastfeeding behaviour and breastfeeding outcome in preterm infants. *Acta Paediatr* 1999;88:1194–203.

58 Pados BF, Estrem HH, Thoyre SM, *et al*. The neonatal eating assessment tool: development and content validation. *Neonatal Netw* 2017;36:359–67.

59 Manea L, Gilbody S, Hewitt C, *et al*. Identifying depression with the PHQ-2: a diagnostic meta-analysis. *J Affect Disord* 2016;203:382–95.

