## [Reviewer comments · BMJ Open]

ARTICLE DETAILS

TITLE (PROVISIONAL)	A mixed-methods, descriptive and observational cohort study examining feeding and growth patterns among low birthweight infants in India, Malawi and Tanzania: the LIFE study protocol
AUTHORS	Vesel, Linda; Spigel, Lauren; Behera, Jnanindra; Bellad, Roopa; Das, Leena; Dhaded, Sangappa; Goudar, Shivaprasad S.; Guruprasad, Gowdar; Misra, Sujata; Panda, Sanghamitra; Shamanur, Latha; Vernekar, Sunil; Hoffman, Irving; Mvalo, Tisungane; Phiri, Melda; Saidi, Friday; Kisenge, Rodrick; Manji, Karim; Salim, Nahya; Somji, Sarah; Sudfeld, CR; Adair, Linda; Caruso, Bethany; Duggan, Christopher; Israel-Ballard, Kiersten; Lee, Anne; Martin, Stephanie; Mansen, Kimberly; North, Krysten; Young, Melissa; Benotti, Emily; Marx Delaney, Megan; Fishman, Eliza; Fleming, Katelyn; Henrich, Natalie; Miller, Kate; Subramanian, Laura; Tuller, Danielle; Semrau, Katherine

VERSION 1 – REVIEW

REVIEWER	Florez, Ivan D University of Antioquia, Pediatrics
REVIEW RETURNED	02-Feb-2021

GENERAL COMMENTS	Authors' manuscript presents the protocol for conducting a Mixed methods research study that aims to document feeding practices and growth patterns among LBW infants in LMIC. This is a nice protocol for a very ambitious project. Authors plan to develop several subprojects each one with its own objectives. My major concern is the methodology, as authors have framed it as a mixed methods' study. The following are my comments: - A mixed methods study, is a specific research design that combines data and analyses from both major approaches (QUAN and QUAL), to respond a mixed methods research (MMR) question (Creswell & Plano-Clark Eds 2017). I do not identify this protocol was written following the major recommendations for a MMR project. Methodologically speaking a MMR requires some points that are not addressed here: 1) Appropriate MMR questions, which include: i) a mixed methods research question; ii) a Quantitative research question, 3) a Qualitative research question. If this is going to be presented as a MMR, authors require to have a specific subheading titled: Research Questions, and this needs to have the three questions, having in mind that the MMR question needs to encompass the QUAN and QUAL questions. Since there are more than one QUAN and QUAL projects, each one should have a question or alternatively, a big overarching
---

	quantitative question might include all the objectives explained in table 1. 2) QUAL projects need to be framed in aore detailed QUAL design. What are the qualitative designs? Are authors interested in Descriptive qualitative analysis, grounded theory, phenomenology, ethnography, case studies? I do not think: "in-depth interviews should be considered as a Design, rather, as a data collection procedure. Thus, a way to deal with this, and clarify information for the other studies, is to add a specific column for "study design". As it is, the second column of Data source, may provide some info for some studies, but do not fully meet the criteria for a study design (e.g., instead of retrospective chart review, should be framed maybe as a cross-sectional study; retrospective chart is just a data collection procedure.). 3) What makes a MMR a unique design, and not just a cluster of mini-studies one after the other, or in parallel, is that a MMR question exists (see Comment one above) and that the studies are clearly connected, either by sequential process in which one informs/supports the other, or in parallel with a clear combination of the results of each other. I cannot clearly identify how these projects inform the other, or "talk each other". I suggest authors clearly frame this MMR in one of the main MMR designs, being the most important: Sequential exploratory or explanatory, Concurrent, Embdded or (the one that I think fits here) the Multiphase (Creswell 2017). 4) Given the complexity of this project, and depending on the design that authors considered described the best this project, A Flow chart or diagram that clearly described each step, and how each study informs the other, not only chronologically but conceptually is needed (Creswell 2017 book, can help with this approach)  - Regarding the populations, although the three countries are LMIC, there are some differences to highlight. Tanzania's neonatal mortality rates are half the ones in India. Also, The fact that authors will gather info in Public hospitals and Private ones, these differences can be surprising, even in LMIC. Namely, procedures, quality, and resources can be substantially different between a private and public hospital in LMIC, with even mortality rates in the former being very low and sometimes even close to rates in High income countries. How authors will handle this (much more in the cohort study, will they adjust for this?) (Note: I acknowledge I am not familiar with Malawi's, India's or Tanzania's environments, it is just my perception from experience in some other LMIC - Cohort study is not clear to me. Is this a Descriptive cohort? Or, are authors planing an analytical cohort studies? if the latter is the case, methods are poorly described What are the exposures and the controls? What are the effect measures, and what statistical approach will be performed, will there be a multivariate analysis to adjust for confounding factors? if the former (Descriptive cohort), this should be explicitly detailed as such - in the QUAN studies, each one should have each own analytic approach. Analyses are written in a very generic way, and it is not clear each study's analysis
--	--

	- in the QUAL study, interviews will be performed in several languages. A specific subsection to deal with all the languages needs to be added to the protocol. Namely, in what languages are the interview going to be performed in each site, should be described, also what authors are going to perform the interviews, and how is the process of translation to English going to be conducted. Are the ones that make the interviews or the analyses, the ones that are going to translate the main quotes/themes to English? Or the analyses will be performed in the original languages and the themes will be created in the original languages and later translated to English, or themes will be created in English?
--	--

REVIEWER	Bazzano, Alessandra Tulane University School of Public Health, Global Community Health and Behavioral Sciences
REVIEW RETURNED	01-Apr-2021

GENERAL COMMENTS	The manuscript describes a mixed method protocol for a descriptive study to set the stage for infant feeding intervention for LBW babies. The authors are to be congratulated for a very complex protocol manuscript which is well organized and thoughtful. Significant attention to detail has resulted in a well-written manuscript providing information on a fairly complex undertaking. 1. However, the study design is not made explicit and it is unclear whether this is a formative research study, an implementation science study, a cross sectional study, a pilot study, etc... And further, the objectives describe assessing acceptability and feasibility, as well as examining predictors which, are not consistent with other descriptions of the study, particularly the title, which is quite confusing for a protocol. 2. The authors have used the SPIRIT guidelines when they are not appropriate as the protocol title describes a descriptive study not an intervention study. This may be the journal requirement however, the authors are suggested to also utilize both the STROBE and COREQ guidelines and checklists--and to cite these in text--in order to ensure that important details have not been omitted in the protocol, particularly related to the qualitative study design and component which is lacking in a number of details that should be included such as theoretical framework 3. E.g. the authors cite implementation science papers on the choice of coding but is this an implementation science study? what is the epistemological orientation), choice of sampling techniques (why convenience rather than purposive? rationale for analysis approach? background of researchers planned to conduct the data collection and analysis etc? These are all important considerations that will impact on the quality of the data for the mixed methods study. 4. Abstract: The bullet point text needs slight revision and clarification. Line 39: "will focus on low birthweight (LBW) infants with birthweights between 1.5kg and 2.5kg since there is limited data on this more predominant group of LBW infants." Please use more appropriate and specific terminology than "more predominant group" or simply state "limited data on this weight group of LBW infants." 5. Line 43 What is the meaning of this text "where manifestations of LBW infants may vary"? Please revise to be more specific/accurate based on evidence, e.g. "where LBW infant's
---

	physiological characteristics may vary", "where Low Birth Weight may have different consequences for nutrition", "where different health conditions are related to LBW" etc... 6. Line 45 add the word "health" before the word facility 7. Please proofread, as there are missing commas throughout and occasional errors in the body of the text, but the reviewer did not have time to pinpoint each one. 8. The Discussion requires additional development, particularly to identify how this study will add to gaps in the literature and what other studies have been conducted in this area, citing study approaches used in the past and how this study approach differs, etc...
--	--

VERSION 1 – AUTHOR RESPONSE

Reviewer #1 - Dr. Ivan D Florez, Universidad de Antioquia, McMaster University

Comment #1: Authors' manuscript presents the protocol for conducting a Mixed methods research study that aims to document feeding practices and growth patterns among LBW infants in LMIC. This is a nice protocol for a very ambitious project. Authors plan to develop several subprojects each one with its own objectives. My major concern is the methodology, as authors have framed it as a mixed methods' study. The following are my comments: A mixed methods study, is a specific research design that combines data and analyses from both major approaches (QUAN and QUAL), to respond a mixed methods research (MMR) question (Creswell & Plano-Clark Eds 2017). I do not identify this protocol was written following the major recommendations for a MMR project. Methodologically speaking a MMR requires some points that are not addressed here

Response #1: Thank you for your comment and for sharing the Creswell 2017 reference; we found it to be a very helpful reference. We have substantially revised our methodology section per Creswell & Plano-Clark and the COREQ and STROBE checklists.

Comment #2: Appropriate MMR questions, which include: i) a mixed methods research question; ii) a Quantitative research question, 3) a Qualitative research question. If this is going to be presented as a MMR, authors require to have a specific subheading titled: Research Questions, and this needs to have the three questions, having in mind that the MMR question needs to encompass the QUAN and QUAL questions. Since there are more than one QUAN and QUAL projects, each one should have a question or alternatively, a big overarching quantitative question might include all the objectives explained in table 1.

Response #2: Thank you for your comment. We have revised the methods and analysis section and replaced Table 1 with a new table that better captures the MMR approach including the specific quantitative, qualitative and mixed-methods (merged quantitative and qualitative data streams) objectives and research questions. We have also revised the language of the objectives to align with the research questions. This is one large project with numerous data collection activities (rather than many projects); we have changed the language to clarify this point.

Comment #3: QUAL projects need to be framed in a more detailed QUAL design. What are the qualitative designs? Are authors interested in Descriptive qualitative analysis, grounded theory, phenomenology, ethnography, case studies? I do not think: "in-depth interviews should be considered as a Design, rather, as a data collection procedure. Thus, a way to deal with this, and clarify

information for the other studies, is to add a specific column for "study design". As it is, the second column of Data source, may provide some info for some studies, but do not fully meet the criteria for a study design (e.g., instead of retrospective chart review, should be framed maybe as a cross-sectional study; retrospective chart is just a data collection procedure).

Response #3: Thank you for your comment. We have revised the methods and analysis section and replaced Table 1 with a new table that clarifies that the qualitative component of LIFE has a descriptive design. Per your suggestion, the revised Table 1 has a row for study design.

Comment #4: What makes a MMR a unique design, and not just a cluster of mini-studies one after the other, or in parallel, is that a MMR question exists (see Comment one above) and that the studies are clearly connected, either by sequential process in which one informs/supports the other, or in parallel with a clear combination of the results of each other. I cannot clearly identify how these projects inform the other, or "talk each other". I suggest authors clearly frame this MMR in one of the main MMR designs, being the most important: Sequential exploratory or explanatory, Concurrent, Embedded or (the one that I think fits here) the Multiphase (Creswell 2017).

Response #4: We have updated the methods and analysis section (see study design and Table 1 in particular) to indicate that we utilized a convergent parallel design.

Comment #5: Given the complexity of this project, and depending on the design that authors considered described the best this project, A Flow chart or diagram that clearly described each step, and how each study informs the other, not only chronologically but conceptually is needed (Creswell 2017 book, can help with this approach)

Response #5: Please see Table 1; we hope it and the edited text better describe the project.

Comment #6: Regarding the populations, although the three countries are LMIC, there are some differences to highlight. Tanzania's neonatal mortality rates are half the ones in India. Also, the fact that authors will gather info in Public hospitals and Private ones, these differences can be surprising, even in LMIC. Namely, procedures, quality, and resources can be substantially different between a private and public hospital in LMIC, with even mortality rates in the former being very low and sometimes even close to rates in High income countries. How authors will handle this (much more in the cohort study, will they adjust for this?) (Note: I acknowledge I am not familiar with Malawi's, India's or Tanzania's environments, it is just my perception from experience in some other LMIC.

Response #6: You are absolutely correct that there will be differences between the sites. This is precisely why we chose them; we want to highlight country- and regional-specific findings. We mentioned in the study setting section that we will document site and regional differences in infant and maternal characteristics, gestational age, birthweight, feeding patterns, and growth. In the data analysis section, we also added that we will stratify analyses and adjust models by site.

Comment #7: Cohort study is not clear to me. Is this a Descriptive cohort? Or, are authors planning an analytical cohort studies? if the latter is the case, methods are poorly described What are the exposures and the controls? What are the effect measures, and what statistical approach will be performed, will there be a multivariate analysis to adjust for confounding factors? if the former (Descriptive cohort), this should be explicitly detailed as such.

Response #7: Yes, this is a descriptive cohort. We have clarified the language in the title, study design, Table 1 and throughout the manuscript.

Comment #8: in the QUAN studies, each one should have each own analytic approach. Analyses are written in a very generic way, and it is not clear each study's analysis

Response #8: Please see the new Table 1 and the revised data management and analysis section.

Comment #9: In the QUAL study, interviews will be performed in several languages. A specific subsection to deal with all the languages needs to be added to the protocol. Namely, in what languages are the interview going to be performed in each site, should be described, also what authors are going to perform the interviews, and how is the process of translation to English going to be conducted. Are the ones that make the interviews or the analyses, the ones that are going to translate the main quotes/themes to English? Or the analyses will be performed in the original languages and the themes will be created in the original languages and later translated to English, or themes will be created in English?

Response #9: We have added information on languages in the data collection section. The data collection will be completed in English or the local language. For quantitative data, all output will be numerical and any free text will be recorded in English. For the qualitative data, IDIs and/or FGDs conducted in the local language will be translated into English when converting to summary tables or to English transcripts.

Reviewer #2 - Dr. Alessandra Bazzano, Tulane University School of Public Health

Comment #1: The manuscript describes a mixed method protocol for a descriptive study to set the stage for infant feeding intervention for LBW babies. The authors are to be congratulated for a very complex protocol manuscript which is well organized and thoughtful. Significant attention to detail has resulted in a well-written manuscript providing information on a fairly complex undertaking.

Response #1: Thank you very much for your comment.

Comment #2: However, the study design is not made explicit and it is unclear whether this is a formative research study, an implementation science study, a cross sectional study, a pilot study, etc....

Response #2: This is formative, descriptive research. We have changed the wording in the title, Table 1 and throughout the manuscript for clarification.

Comment #3: And further, the objectives describe assessing acceptability and feasibility, as well as examining predictors which, are not consistent with other descriptions of the study, particularly the title, which is quite confusing for a protocol

Response #3: We have reframed the objectives and the description of the study to foster better clarity. Please see the study design section and Table 1. We agree that the original words used were misleading.

Comment #4: The authors have used the SPIRIT guidelines when they are not appropriate as the protocol title describes a descriptive study not an intervention study. This may the journal requirement however, the authors are suggested to also utilize both the STROBE and COREQ guidelines and

checklists--and to cite these in text--in order to ensure that important details have not been omitted in the protocol, particularly related to the qualitative study design and component which is lacking in a number of details that should be included such as theoretical framework.

Response #4: We have edited the manuscript using STROBE and COREQ guidelines and checklists and have cited them in the study design section of the manuscript. We have included many more details throughout the manuscript related to design, data collection and data analysis.

Comment #5: E.g. the authors cite implementation science papers on the choice of coding but is this an implementation science study? what is the epistemological orientation), choice of sampling techniques (why convenience rather than purposive? rationale for analysis approach? background of researchers planned to conduct the data collection and analysis etc? These are all important considerations that will impact on the quality of the data for the mixed methods study.

Response #5: Thank you for the comment; we have clarified the manuscript to note this as a descriptive observational cohort study. We have added details in the data collection section on qualitative sampling and in the data analysis section on analysis and background of analysts.

Comment #6: Abstract: The bullet point text needs slight revision and clarification. Line 39: "will focus on low birthweight (LBW) infants with birthweights between 1.5kg and 2.5kg since there is limited data on this more predominant group of LBW infants." Please use more appropriate and specific terminology than "more predominant group" or simply state "limited data on this weight group of LBW infants."

Response #6: We have changed the sentence per your suggestion.

Comment #7: Line 43 What is the meaning of this text "where manifestations of LBW infants may vary"? Please revise to be more specific/accurate based on evidence, e.g. "where LBW infant's physiological characteristics may vary", "where Low Birth Weight may have different consequences for nutrition", "where different health conditions are related to LBW" etc...

Response #7: We have changed "manifestations" to phenotypes as we are referring to the fact that in sub-Saharan Africa, LBW infants are predominantly preterm (either small-for-gestational age or appropriate-for-gestational age) while in South Asia they are predominantly term small-for-gestational age.

Comment #8: Line 45 add the word "health" before the word facility

Response #8: The word "health" has been added.

Comment #9: Please proofread, as there are missing commas throughout and occasional errors in the body of the text, but the reviewer did not have time to pinpoint each one.

Response #9: We have carefully proofread the entire document.

Comment #10: The Discussion requires additional development, particularly to identify how this study will add to gaps in the literature and what other studies have been conducted in this area, citing study approaches used in the past and how this study approach differs, etc...

Response #10: Thank you for this comment. We have revised the discussion section to include gaps in the literature and the added value of this work.

VERSION 2 – REVIEW

REVIEWER	Florez, Ivan D University of Antioquia, Pediatrics
REVIEW RETURNED	04-Jul-2021

GENERAL COMMENTS	Authors have revised the manuscript according to my comments. I am happy with the revisions. The methodology seems clear now, and all the limitations I highlighted are now addressed.
--

REVIEWER	Bazzano, Alessandra Tulane University School of Public Health, Global Community Health and Behavioral Sciences
REVIEW RETURNED	08-Oct-2021

GENERAL COMMENTS	The authors have done well in responding and revising based on reviewer comments. There are only a few minor points, which also may benefit from Journal/Editorial input.  1) The authors have changed the tense throughout from future to present, presumably because they had to begin their work despite review timelines. It is not clear to this reviewer whether that is appropriate from an editorial pov for a Protocol manuscript--the BMJ editorial policies may be consulted on this point. 2) The sentence in the introduction "Generally studies capturing evidence on nutritional interventions... are lacking" should be revised. It is not clear in what way they are lacking (in number? in quality?) and the reference provided is a review of reviews--which would indicate there is a sufficient quantity of research to review--specifically on milk type and fortification rather than on what the authors describe as "nutritional interventions" more generally that could include other approaches. 3) Table 1 could be made more succinct and readable, perhaps by improving the formatting or removing the column labeled "Strands" which does not describe the content with accuracy. 4) Near the end of the Methods and Analysis section, the section for "Qualitative" Data Management and Analysis is woefully short and missing details included in the COREQ checklist which has been referenced. Please add detail and reconcile the description of inductive analysis procedures with subsequent naming of "A comprehensive deductive analysis of the data". From the reviewers read, it appears that a mix of inductive and deductive analysis will be used, so that can be described along with more detail from COREQ which are missing. 5) It is not clear why the Discussion section first paragraph mentions the following "Although human milk is promoted by the WHO...exclusive breastfeeding alone may not provide" What is the rationale for including this statement? It seems to appear without
---

	any clear link to the protocol. Also it leaves out the possibility of human milk based fortifiers? 6) The authors should consider de-colonizing and inclusive principles in the Manuscript. For example are the "community stakeholders" included as co-authors? What role will they have beyond being consulted? See also the language of "buy-in" in the Acknowledgements. The authors may also wish to note gender inclusive language is available for breast/chestfeeding and birthing parents (see Bartick, Stehel, et al. Academy of Breastfeeding Medicine 2021).
--	--

VERSION 2 – AUTHOR RESPONSE

Reviewer #1 - Dr. Ivan D Florez, Universidad de Antioquia, McMaster University

Comment #1: Authors have revised the manuscript according to my comments. I am happy with the revisions. The methodology seems clear now, and all the limitations I highlighted are now addressed.

Response #1: Thank you again for your suggestions. We are pleased to hear that you are happy with our revisions.

Reviewer #2 - Dr. Alessandra Bazzano, Tulane University School of Public Health

Comment #1: The authors have done well in responding and revising based on reviewer comments. There are only a few minor points, which also may benefit from Journal/Editorial input. The authors have changed the tense throughout from future to present, presumably because they had to begin their work despite review timelines. It is not clear to this reviewer whether that is appropriate from an editorial point for a Protocol manuscript--the BMJ editorial policies may be consulted on this point.

Response #1: You are absolutely correct that we used present tense throughout because some of this work is ongoing since we submitted the manuscript for review in December 2020. The author guidelines posted on the BMJ Open website state that "protocol manuscripts should report planned or **ongoing** research studies." Therefore, we have retained the present tense throughout the manuscript.

Comment #2: The sentence in the introduction "Generally studies capturing evidence on nutritional interventions... are lacking" should be revised. It is not clear in what way they are lacking (in number? in quality?) and the reference provided is a review of reviews--which would indicate there is a sufficient quantity of research to review--specifically on milk type and fortification rather than on what the authors describe as "nutritional interventions" more generally that could include other approaches.

Response #2: We revised the sentence in the manuscript to the following: "Generally, studies capturing evidence on optimal milk type and content for moderately LBW infants (1.5kg to <2.5kg) in low-resource settings are lacking, hindering the proper management of nutritionally at-risk newborns (18)."

Comment #3: Table 1 could be made more succinct and readable, perhaps by improving the

formatting or removing the column labeled "Strands" which does not describe the content with accuracy.

Response #3: We have formatted Table 1 and included the relevant section per the recommendation of reviewer #1 based on the mixed-methods approach described by Creswell & Plano-Clark Eds 2017. We adjusted the column widths to be similar and updated formatting to be consistent. We are happy to make additional changes based on specific recommendations from the Editor/Copy-editor at BMJ Open if accepted for publication.

Comment #4: Near the end of the Methods and Analysis section, the section for "Qualitative" Data Management and Analysis is woefully short and missing details included in the COREQ checklist which has been referenced. Please add detail and reconcile the description of inductive analysis procedures with subsequent naming of "A comprehensive deductive analysis of the data". From the reviewers read, it appears that a mix of inductive and deductive analysis will be used, so that can be described along with more detail from COREQ which are missing.

Response #4: Thank you for your comments. We have revised the relevant paragraph. We have also updated the COREQ to (a) indicate that certain items are "N/A" because the manuscript is a protocol and (b) to replace some of the items previously noted as "N/A" with a page reference to the manuscript with edited text.

Comment #5: It is not clear why the Discussion section first paragraph mentions the following "Although human milk is promoted by the WHO...exclusive breastfeeding alone may not provide" What is the rationale for including this statement? It seems to appear without any clear link to the protocol. Also it leaves out the possibility of human milk based fortifiers?

Response #5: We have removed this sentence since we agree it is not needed here.

Comment #6: The authors should consider de-colonizing and inclusive principles in the Manuscript. For example are the "community stakeholders" included as co-authors? What role will they have beyond being consulted? See also the language of "buy-in" in the Acknowledgements. The authors may also wish to note gender inclusive language is available for breast/chestfeeding and birthing parents (see Bartick, Stehel, et al. Academy of Breastfeeding Medicine 2021).

Response #6:

Thank you for highlighting these important points. Our consortium consists of 10 organizations, and more than 50 team members in India, Malawi, Tanzania and the United States. Through a publications committee consisting of study PIs from India, Malawi, Tanzania and the United States, we developed an inclusive and representative writing process, determination of authorship, and strategy to provide writing opportunities for early career team members. We have incorporated the ICJME criteria and defined clear principles of diversity, equity and inclusion. With nearly 40 authors on this paper, representing data collectors, researchers and clinicians, we believe that we have attempted to address de-colonizing and inclusive principles. In the acknowledgements, we have changed "buy-in" to "partnership." Thank you for raising the important concept of gender inclusive language and for sharing the reference. We recognize the importance of inclusive language and its implications on dignity and equality. For the communities with whom we work as part of the Low Birthweight Infant Feeding Exploration (LIFE) study, the terms that have been used are woman,

mother, maternal and breastfeeding. In our manuscript, we used language that is most appropriate for the contexts in which the research is being conducted.